# Recent Advances in Optimizing Radiation Therapy Decisions in Early Invasive Breast Cancer

**DOI:** 10.3390/cancers15041260

**Published:** 2023-02-16

**Authors:** Nazia Riaz, Tiffany Jeen, Timothy J. Whelan, Torsten O. Nielsen

**Affiliations:** 1Department of Pathology and Laboratory Medicine, University of British Columbia, Vancouver, BC V6T 1Z4, Canada; 2Department of Oncology, McMaster University, Hamilton, ON L8S 4L8, Canada; 3Division of Radiation Oncology, Juravinski Cancer Centre at Hamilton Health Sciences, Hamilton, ON L8V 5C2, Canada

**Keywords:** breast cancer, breast-conserving surgery, radiation, de-escalation, biomarkers, prognosis

## Abstract

**Simple Summary:**

Radiation therapy is routinely prescribed for women who undergo breast-sparing surgery for early breast cancers. Over the years, advancements in diagnosis and treatments have dramatically improved breast cancer outcomes, now approaching 100% survival at 5 years for those diagnosed at stage I with favorable clinical and molecular features. In this review, we discuss the investigations that are underway to identify women with low-risk cancers in whom radiation therapy can either be completely avoided or delivered in lower intensities. We also review ongoing clinical trials that are assessing if radiation therapy can increase the capacity of patients’ anticancer immune responses and discuss if cancer cells that are shed in the blood can guide radiation decisions.

**Abstract:**

Adjuvant whole breast irradiation after breast-conserving surgery is a well-established treatment standard for early invasive breast cancer. Screening, early diagnosis, refinement in surgical techniques, the knowledge of new and specific molecular prognostic factors, and now the standard use of more effective neo/adjuvant systemic therapies have proven instrumental in reducing the rates of locoregional relapses. This underscores the need for reliably identifying women with such low-risk disease burdens in whom elimination of radiation from the treatment plan would not compromise oncological safety. This review summarizes the current evidence for radiation de-intensification strategies and details ongoing prospective clinical trials investigating the omission of adjuvant whole breast irradiation in molecularly defined low-risk breast cancers and related evidence supporting the potential for radiation de-escalation in HER2+ and triple-negative clinical subtypes. Furthermore, we discuss the current evidence for the de-escalation of regional nodal irradiation after neoadjuvant chemotherapy. Finally, we also detail the current knowledge of the clinical value of stromal tumor-infiltrating lymphocytes and liquid-based biomarkers as prognostic factors for locoregional relapse.

## 1. Introduction

Breast-conserving surgery consolidated with adjuvant radiation has been the well-established standard of care for women diagnosed with early invasive breast cancer for more than three decades [1,2,3,4]. The Early Breast Cancer Trialists’ Collaborative Group patient-level meta-analysis of 17 randomized trials, including more than 10,000 women, provides compelling evidence favoring adjuvant radiotherapy over no radiotherapy after breast-conserving surgery for a 10-year absolute risk reduction of 15.7% for any recurrence and a moderate absolute decrease in breast cancer mortality by 3.8% at 15 years [5]. Supplementation with a tumor bed radiation boost further diminishes the relative risk of local recurrence by 50% in high-risk patients [6]. Over the past several decades, screening, early diagnosis, refinements in imaging, surgical techniques, pathological evaluation, improved understanding of tumor biology, and now the standard use of more effective neo/adjuvant systemic therapies have contributed to a steady and substantial improvement in clinical outcomes [7].

Modern radiotherapy techniques incorporating hypofractionation schedules have improved quality of life, decreased hospital stay, and lessened side effects compared to traditional radiotherapy modalities for early breast cancer [8,9]. Furthermore, despite a lack of proven survival benefits in some instances, optimal locoregional control undoubtedly contributes to improved quality of life [8].

Similar to systemic therapy decisions, radiotherapy should be evaluated using an individualized approach to avoid over-treatment for early invasive breast cancer. This need has prompted re-consideration of radiotherapy indications and has initiated investigations to identify any subset of low-risk women with such a negligible burden of residual locoregional disease risk following breast-conserving surgery who could be safely spared radiation therapy. In particular, much attention has been focused lately on elderly early-stage breast cancer patients with favorable prognostic factors [10].

Among these different studies, retrospectively analyzed data by Herskovic et al. has shown an improvement in overall survival with adjuvant radiation in a cohort of >60,000 women > 65 years of age [11]. In contrast, a series of at least seven first-generation clinical trials were conducted between 1981–1998 that evaluated rates for local recurrences and overall survival after breast-conserving surgery with or without radiation. These trials limited the eligibility criteria for patient enrollment to T1–2 node-negative cancers with microscopically clear resection margin status. While no survival benefit was observed, a sufficiently low-risk group in terms of local recurrence could not be identified, signifying the insufficient capacity of standard clinicopathological factors alone in this regard [12]. This, perhaps, was also compounded by less established standards for hormone receptor assessment and variability in the definition of pathologically clear surgical margins at that time.

Among the second generation of relevant phase III clinical trials, noteworthy are the Cancer and Leukemia Group B 9343 cooperative group (CALGB 9343) and the Postoperative Radiotherapy in Minimum-risk Elderly (PRIME II) trials that exercised more stringent eligibility criteria based on age at diagnosis (≥70 years and ≥65 years for CALGB 9343 and PRIME II, respectively) and favorable tumor characteristics. For the CALGB 9343 trial, at 10 years, 98% of women randomized to tamoxifen and radiation after breast-conserving surgery, versus 90% of those in the tamoxifen-only arm, remained free from local and regional relapses [13]. PRIME II yielded comparable results at 10 years, showing an ipsilateral breast tumor recurrence rate of 0.9% with and 9.8% without radiation [14,15]. Even with this statistically significant improvement in the risk of locoregional relapses with radiation in both these trials, no prominent impact was noted on survival. The level I evidence thus generated led to a modification in the clinical practice guidelines allowing radiation omission after breast-conserving surgery in women ≥ 70 years with T1N0, hormone receptor-positive early breast cancers who are committed to complete a 5-year course of endocrine therapy [16] as low compliance with endocrine therapy is associated with poor locoregional control when radiation therapy is also being omitted from the treatment plan [17].

Regardless of these recommendations, the use of radiation therapy has continued among elderly women, the decision largely influenced by patients’ age and physicians’ preference [18,19]. Additionally, achieving higher locoregional control with radiation may be the preferred choice of women to avoid the deterioration in quality of life and the financial costs associated with locoregional recurrence, particularly in the presence of poor prognostic factors such as grade 3 histology and positive surgical margins. It has also been reported that elderly women may prefer to receive radiation therapy (that is delivered over weeks) over adjuvant endocrine therapy (delivered over 5 years, with often poor compliance outside clinical trial settings) [17].

Beyond the omission of adjuvant radiation in select indolent tumors in elderly patients, de-intensification strategies have continued to evolve over the past two decades and have positively contributed towards patient convenience and compliance by reducing radiation duration and toxicities without compromising oncological safety.

Despite its undeniable benefits, radiation therapy is linked with the risk of significant morbidities. Radiation dermatitis is the most common early complication of adjuvant radiation following breast-conserving surgery that, if severe, may potentially interrupt the radiation schedule [20,21]. Furthermore, while the risk of acute radiation toxicity is significantly lower with partial breast radiation compared to whole breast radiation, the risk of delayed dermal toxicities, including telangiectasia, fat necrosis, and subcutaneous fibrosis, has been shown to be increased in some studies [22]. Additionally, both early and delayed arm lymphedema remains a debilitating morbidity occurring in every fifth breast cancer survivor, negatively impacting their quality of life and associated with an increased burden on the health care system [23]. In particular, regional radiation therapy is a risk factor contributing to the development of late-onset lymphedema (>12 months) [24] and cardiac and pulmonary complications [25]. However, at least with regard to cardiac toxicity, the use of CT-based radiotherapy planning greatly mitigates this risk [26,27]. Lastly, despite encouraging response rates to pre-operative radiation in early-stage breast cancer, wound-related complications remain a major concern that has prompted further investigations to optimize radiation doses and schedules [28].

In this review, we will summarize the recent advances in hypofractionated and accelerated partial breast irradiation and discuss the ongoing clinical trials that utilize existing validated genomic classifiers and immunohistochemistry-based assays for risk categorization and radiation omission in hormone receptor-positive early breast cancers. We will also provide an overview of the recent literature supporting the potential for radiation de-escalation in HER2-positive and triple-negative subtypes. Next, we will discuss clinical trials underway for the de-escalation of regional nodal radiation after neoadjuvant chemotherapy. Lastly, we will provide an update on current advances in the utilization of immune biomarkers (specifically stromal lymphocytes) and liquid-biopsy-based approaches for prognostication of locoregional risk and prediction of radiation benefit.

## 2. De-Intensification of Radiation in Early Breast Cancer: Hypofractionation and Accelerated Partial Breast Radiation

Until about a decade ago, the conventional dosage for whole breast irradiation, defined as 45–50 Gy given in 25 fractions of 1.8–2.0 Gy once a day over 5 weeks, with or without a tumor bed boost (suggested as 10 Gy in 4–5 fractions) had been the standard of care [29]. This extended duration of treatment has been associated with acute and late radiation-induced toxicities, poor quality of life, low compliance, increased workload, and high costs incurred by healthcare systems [30,31]. In addition, several factors, including patient age, co-morbidities, income, ethnicity, education attainment, distance to the treatment facility, and the availability of radiation oncologists, have been found to be associated with disparities in the receipt of radiotherapy [32,33,34,35]. These barriers contribute to an increased rate of mastectomy among women who might otherwise have chosen breast-conserving surgery with adjuvant radiation [36].

Contrary to the aforementioned conventional radiation, in the hypofractionation approach, a higher dose (>2 Gy) per fraction is delivered in fewer fractions over a shorter duration, such that a lower overall total dose is delivered. The radiobiologic rationale for hypofractionation is based on the concept of fractionation sensitivity (α/β therapeutic ratio) such that if the fractionation sensitivity of the cancer cells is similar to the fractionation sensitivity of irradiated normal cells, a higher dose per fraction can be delivered to achieve tumoricidal effect while limiting toxicities to the normal breast [37].

High-quality, mature follow-up data from multiple phase III randomized clinical trials have consistently demonstrated non-inferiority of moderate hypofractionated radiation (40–42.5 Gy in 15–16 fractions of 2.6–2.7 Gy over 3 weeks) compared to conventional whole breast irradiation, for improving locoregional control, overall survival, and cosmetic outcomes while reducing normal tissue toxicities [38,39,40,41]. Based on the level I evidence generated from these clinical trials, guidelines recommend hypofractionation as the standard of care [29]. As a further radiation de-intensification strategy, more recently, the 5-year results from the FAST FORWARD trial have shown the non-inferiority of an ultra-hypofractionation regimen (26 Gy in 5 fractions of 5.2 Gy over 1 week versus standard hypofractionation of 40 Gy in 15 fractions over 3 weeks) for local control of the conserved breast or chest wall without compromising normal breast tissue [42].

Given that most in-breast recurrences occur in the index quadrant [43], Accelerated Partial Breast Irradiation (APBI) is an alternative approach to hypofractionation that delivers targeted radiation (>2 Gy) to the lumpectomy site (and the associated margin) over a period of 2–5 days and has shown promising results for oncological safety and cosmetic outcomes, while decreasing the treatment time to 2–5 days. Methods of APBI delivery include single or multiple catheter brachytherapy [44,45,46,47,48], intraoperative radiotherapy [49,50,51], and the use of external beam radiation therapy techniques such as three-dimensional conformal radiation therapy [42,52,53,54] and intensity-modulated radiation therapy [55,56].

One caveat to the use of APBI is the careful selection of eligible patients. This is particularly evident from the phase III NSABP B-39/RTOG 0413 trial, which was unable to demonstrate non-inferiority of APBI compared to whole breast irradiation with regard to the ipsilateral breast tumor recurrence rate. This is likely because the trial included a heterogenous population comprising low-risk as well as high-risk patients with features such as age < 40, invasive lobular carcinomas, multifocality, tumor size > 2 cm, 1–3 positive lymph nodes, or hormone receptor-negative status [57]. However, the absolute difference (1%) in ipsilateral breast tumor recurrence was found to be small and potentially acceptable to some patients.

Variations in APBI fraction size, delivery methods, and radiation schedules are associated with different cosmesis and tissue toxicity results. On the one hand, ABPI (compared to whole breast irradiation) has shown comparable or improved cosmetic outcomes and toxicity profiles [58,59,60,61], while on the other hand, some trials have shown contrary results. For example, in the OCOG-RAPID trial, compared to whole breast irradiation, APBI delivered via three-dimensional conformal radiation therapy as 38.5 Gy in 10 fractions twice per day over 5–8 days was associated with a higher rate of delayed radiation toxicity and poor cosmesis [53]. The authors noted that this could be the result of short dosing intervals (daily doses separated by 6–8 h) and that the worse cosmesis could be potentially circumvented by a once-daily dosing regimen. Likewise, the physician reported cosmetic outcomes at 3 years were inferior with APBI compared to whole breast irradiation in NSABP B-39/RTOG 0413, which used a similar fractionation schedule [57].

Intraoperative radiotherapy that delivers a single fraction with electrons or soft X-rays intraoperatively immediately after tumor resection is yet another strategy to decrease the radiotherapy time and has been evaluated in two randomized controlled trials. The European Institute of Oncology’s ELIOT trial showed increased local and regional relapse rates associated with intraoperative radiotherapy compared to conventional whole breast radiation at a median follow-up of 12.4 years (11% versus 2%), despite there being no significant difference in the overall survival rate between the two groups. The exploratory analysis identified several factors associated with a significantly increased risk of ipsilateral breast tumor recurrence: tumor size > 2 cm, grade 3, ≥4 positive axillary nodes, Ki-67 > 20%, and luminal B or triple-negative clinical subtype [51]. However, intraoperative radiotherapy as administered in ELIOT may still be an appropriate option for a subset of patients with extremely favorable tumor biology (well-differentiated luminal A cancers < 1 cm in size with Ki-67 < 14%) who experienced a 10-year ipsilateral breast tumor recurrence rate of <1.3% [51]. The second phase III TARGIT-IORT trial found intraoperative radiotherapy to be non-inferior to whole breast radiation in terms of five-year oncological outcomes. In particular, one stratum of the trial was designed to include a risk-adapted approach whereby patients receiving intraoperative radiotherapy, if found to have high-risk features on final pathology, would then receive standard whole breast radiation post-operatively; in those cases, the intraoperative dose was considered as a tumor bed boost [62]. Further detailed analyses have shown the oncological safety of the TARGIT-IORT risk-adapted approach in all relevant subgroups stratified by breast cancer subtype, nodal involvement, tumor size, and grade. Intriguingly, the results also suggest that contrary to standard whole breast radiation, local failures occurring in the TARGIT-IOTR arm are not necessarily associated with poor survival. While the biological mechanisms are not completely understood, this effect could be partly explained by an abscopal effect of intraoperative radiotherapy delivered to a well-vascularized tumor bed [63]. Of note, in the second TARGIT stratum, delayed intraoperative radiotherapy delivered at a median of 37 days at a second surgery failed to demonstrate non-inferiority to standard whole breast irradiation [64]. Although the TARGIT-IORT procedure has been incorporated into clinical practice globally [65], by and large, the intraoperative radiotherapy approach is still regarded as investigational until mature, long-term data becomes available [66,67].

A recently published meta-analysis of 15 clinical trials, including more than 16,000 patients, compared partial breast radiation with whole breast radiation and reported the rates of any ipsilateral breast tumor recurrences as the primary outcome measure. Collectively, partial breast radiation was associated with a higher risk of ipsilateral breast recurrences compared to whole breast radiation (5% versus 2.8%). Of note, after excluding intraoperative radiotherapy trials, the rates of ipsilateral recurrences were 3.3% with partial breast radiation versus 2.6% with whole breast radiation. Another noteworthy observation from this meta-analysis is the higher rate of elsewhere recurrence in the ipsilateral breast with partial breast radiation compared to whole breast radiation. The rates for true/marginal recurrence were, however, comparable between the two treatment modalities. Despite the advantage over whole breast radiation in limiting the risk of acute toxicities, overall partial breast radiation yielded inferior effectiveness [68]. However, these results varied with the delivery techniques such that multi-catheter brachytherapy and external beam radiation approaches with CT planning were associated with higher oncological safety compared to the intraoperative approach.

## 3. Ongoing Clinical Trials for Guiding Adjuvant Radiation Omission Decisions in Women with Hormone Receptor-Positive Early-Stage Breast Cancers

The clinical utility of validated genomic and immunohistochemistry-based biomarkers for guiding adjuvant radiation omission decisions following breast-conserving surgery in favorable risk invasive breast cancers is under intense prospective investigation (Table 1). The common theme of these third-generation clinical trials is to combine clinical factors with some type of molecular risk assay to identify a low-risk group whose prognosis is so good, at least in the context of adequate endocrine therapy, that radiation could not provide significant additional benefit.

Amongst these trials, the prespecified 5-year interim analysis of the LUMINA prospective trial was the first to be presented at the 2022 American Society of Clinical Oncology meeting [69]. Briefly, LUMINA is a multicentre, single-arm prospective cohort study investigating the clinical value of clinicopathological characteristics together with Ki-67 immunohistochemistry-based phenotyping for identifying women ≥ 55 years with sufficiently low-risk molecularly defined T1N0 luminal A breast cancers (ER ≥ 1%, PR > 20%, HER2 negative, and Ki-67 ≤ 13.25%) who can be adequately treated with breast-conserving surgery and endocrine therapy alone without compromising oncological outcomes. The primary endpoint is the ipsilateral local recurrence of any invasive or non-invasive breast cancer. Amongst the 501 enrolled patients, the reported 5-year local recurrence rate of 2.3% (90% CI 1.3–3.8) was well below the prespecified boundary of significance (5%), making this a positive study. Moreover, the 5-year rates for contralateral breast cancer, relapse-free, disease-free, and overall survival are 1.9% (90% CI 1.1–3.2), 97.3% (90% CI 95.9–98.4), 89.9% (90% CI 87.5–92.2) and 97.2% (90% CI 95.9–98.4), respectively. While the full analysis is awaited, these 5-year results do provide prospective data supporting the safe omission of adjuvant whole breast radiation in precisely selected luminal A breast cancers with a low Ki-67 index (≤13.25%), quantified using a standardized, validated, decentralized IHC assay [77].

PRECISION (Profiling Early Breast Cancer for Radiotherapy Omission) is a phase II, single-arm prospective cohort study led by the Dana-Faber Cancer Institute that aims to evaluate the 5-year risk of ipsilateral locoregional recurrence following upfront breast-conserving surgery without whole breast radiation. Enrollment criteria comprise women aged 50–75 years with ER+/PR+/HER2−, pT1N0M0, grade 1–2 invasive breast cancers. The tumors from the eligible patients are subjected to central PAM50 transcriptional profiling using the Prosigna assay. Only women whose tumors yield a low risk of Recurrence (ROR) score corresponding to the luminal A subtype qualify to forego radiation to the conserved breast and are offered 5 years of endocrine therapy only [70,78]. At a median follow-up of 26.9 months, 12 events have been recorded among 382 women with a ROR ≤ 40 (4, ipsilateral in-breast recurrences; 7, contralateral breast cancers; and 1, unrelated melanoma). No regional-nodal or distant recurrences have been reported thus far. The 2-year cumulative rate of locoregional recurrence is 0.3% (95% CI: 0–1.0%) [79].

IDEA (Individualized Decisions for Endocrine Therapy Alone) is an American, multicentre, single-arm prospective cohort study headed by the University of Michigan Rogel Cancer Centre, enrolling postmenopausal women (50–69 years) with ER+/PR+/HER2−, unifocal, pT1N0M0 breast cancer. This study aims to determine if 5-year locoregional relapse risk remains sufficiently low after breast-conserving surgery and 5 years of endocrine therapy (tamoxifen or aromatase inhibitor) when radiation therapy is withheld from the treatment plan. The characterization of genomically low-risk tumors is based on an Oncotype Dx recurrence score ≤ 18 [71].

PRIMETIME is a multicentre UK-based prospective case-cohort study that is evaluating if adjuvant radiation can be safely avoided in very low-risk women ≤ 60 years surgically treated with breast conservation followed by standard endocrine therapy. The study’s inclusion criteria with regards to clinicopathological tumor characteristics are similar to the PRECISION trial, but the molecular risk eligibility will incorporate the very low-risk category assessed using a validated immunohistochemistry-based prognostic algorithm called IHC4+Clinical (IHC4+C). This recurrence probability score combines the protein expression of triple receptors and Ki-67, along with a Clinical Treatment Score (age, tumor size, nodal status, tumor grade, and endocrine treatment: tamoxifen versus anastrozole) to stratify the residual disease risk into four categories: very low, low, intermediate and high [80,81]. Women whose tumors classify as very low risk qualify for enrollment in PRIMETIME. The primary endpoint is 5-year ipsilateral breast tumor recurrence [72].

De-escalation of Breast Radiation (DEBRA-NRG BR007) is an NRG Oncology-sponsored, multicentre phase III clinical trial that is investigating whether breast-conserving surgery followed by endocrine therapy is non-inferior to breast-conserving surgery followed by endocrine therapy and standard whole breast irradiation. Eligibility criteria include patients aged 50–70 years who are diagnosed with unicentric ER+/PR+/HER2− pT1N0 breast cancer that are also genomically characterized as low-risk by an Oncotype Dx Recurrence Score of ≤ 18. The primary endpoint is invasive or non-invasive in-breast tumor recurrence. The trial is currently recruiting and will accrue 1714 patients enabling a final randomization cohort of 1670 patients (835 per arm). As of 30 June 2022, 169 patients had been screened and 147 randomized [73,82].

The EXPERT trial (EXamining PErsonalised Radiation Therapy for low-risk early breast cancer) is an initiative of Breast Cancer Trials in Australia and New Zealand that also uses PAM50/Prosigna for molecular risk stratification. EXPERT is a randomized phase III trial of adjuvant radiation versus observation, following breast-conserving surgery and endocrine therapy for molecularly defined luminal A breast cancer with a low ROR score (≤60) amongst pre or postmenopausal women ≥ 50 years. The clinicopathological factors deemed necessary for inclusion are similar to the PRECISION trial. The primary endpoint is local recurrence at 10 years [74].

The DBCG-RT Natural trial, sponsored by the Danish Breast Cancer Cooperative Group, is a non-inferiority phase III clinical trial designed to compare the 5-year risk of local recurrence between partial breast irradiation versus no irradiation among women ≥60 years with unifocal, pT1N0M0, ER+/HER2− invasive ductal carcinomas treated with breast-conserving surgery. This is the only radiation de-escalation trial in which low-risk patient selection is ascertained purely by traditional clinicopathological features [75].

While the above studies are specifically designed for evaluating the safety of omitting whole breast radiation in stage I node-negative breast cancers, the Canadian Cancer Trials Group MA.39 TAILOR RT is a phase III biomarker-directed randomized trial designed to test the non-inferiority of omitting regional nodal irradiation versus regional lymph node irradiation post-lumpectomy and omitting locoregional radiotherapy to the chest wall and regional nodes versus locoregional radiotherapy following mastectomy in women ≥ 35 years. All patients will receive endocrine therapy. The eligibility criteria include ER+, HER2− breast cancers with 1–3 positive axillary lymph nodes, and an Oncotype DX Recurrence score ≤ 25. The primary endpoint will measure any recurrence or death due to breast cancer [76].

## 4. Evidence for Radiation De-Escalation in HER2-Positive and Triple-Negative Breast Cancers

### 4.1. HER2+ Early Breast Cancer

About 15–20% of women are diagnosed with HER2+ early breast cancers. The Early Breast Cancer Trialists’ Collaborative Group’s patient-level meta-analysis of seven randomized clinical trials, including 13,864 women, has confirmed the benefit of adjuvant trastuzumab to chemotherapy in reducing the risk of any invasive breast cancer recurrence and breast cancer-specific mortality by a third in operable breast cancers regardless of the nodal status. In addition, the risk of the first isolated local recurrence was also reduced significantly with trastuzumab treatment [83].

While currently there are no completed clinical trials of radiation omission in HER2+ breast cancer, some insights are gained from observational studies that highlight the value of HER2-targeted therapies in achieving good locoregional control. For instance, Bazan et al. performed a retrospective analysis using the National Cancer Database and identified a cohort of T1N0 HER2+ patients treated with breast conservation, adjuvant chemotherapy, and HER2-targeted therapies. Of these, 6388 patients were treated with adjuvant radiation, while 509 were radiation naïve. Patients in the radiation naïve group experienced a significantly inferior 2-year overall survival compared to those who received adjuvant radiation (88.9% versus 99.2%, respectively). The study has several limitations, including a lack of information on locoregional relapses or cancer-specific survival, short follow-up, and, importantly, non-compliance with systemic therapies that may have contributed to an exaggerated poor overall survival in the radiation naïve group [84].

Some recent trials have investigated the de-intensification of chemotherapies and HER2-targeted therapies in early-stage HER2+ breast cancer, where all patients received radiation as per standard protocol. Encouraging results of these trials highlight the effectiveness of chemotherapies and HER2-targeted therapies for improving the clinical outcomes, including lowering the risk of locoregional relapses. Amongst these, the single-arm, multicentre phase II Adjuvant Paclitaxel and Trastuzumab (APT) trial included patients with T1–2N0–N1mic HER2+ breast cancer treated with upfront surgery followed by radiation therapy (for breast-conserving surgery only). All patients received adjuvant paclitaxel with trastuzumab for 12 weeks, followed by the continuation of trastuzumab for 1 year [85,86]. Only 5 of the 406 patients developed locoregional recurrences, resulting in an impressive 7-year locoregional recurrence-free survival (with radiotherapy) of 98.6% (95% CI 97.4–99.8%) [87].

The phase II ATEMPT trial randomized women with HER2+ T1–2N0–N1mic breast cancer to adjuvant trastuzumab emtansine (T-DM1) versus paclitaxel plus trastuzumab to investigate if the two treatments had comparable efficacy and toxicity profile [88]. A subsequent retrospective-prospective analysis reported at 3 years showed an extremely low rate of isolated local recurrences such that only 2 events were recorded in the group treated with T-DM1 (*n* = 383) and 4 in the paclitaxel plus trastuzumab arm (*n* = 114), though it should be noted that the inclusion criteria specified that participants who underwent breast-conserving surgery were required to receive radiation therapy and those who underwent mastectomy were permitted to receive radiation therapy to the chest wall and regional lymph nodes [89]. In the KATHERINE trial, high-risk HER2+ breast cancer patients with residual invasive disease following neoadjuvant chemotherapy and HER2-targeted therapy were randomized to adjuvant T-DM1 versus trastuzumab. Patients received adjuvant radiation therapy as per participating institutional guidelines. The trial yielded positive results showing an impressive 50% relative reduction in the risk of invasive recurrence or mortality favoring the use of T-DM1 [90], leading to subsequent FDA approval [91,92]. Overall, a very low rate of locoregional recurrences was recorded in both the treatment arms (trastuzumab group, *n* = 743: 4.6%; T-DM1 group, *n* = 743: 1.1%) in patients who were HER2+ in pre-treatment biopsies but tested negative on the residual disease biopsy [93]. Albeit long-term follow-up from these trials is warranted, the substantially low risk of locoregional relapses is indeed encouraging and may pave the way for future clinical trials investigating radiotherapy de-escalation strategies in stringently selected low-risk early-stage HER2+ breast cancers. In this trial, neither radiation modalities nor the sequence of integration of systemic treatment with radiotherapy are specified. This information could be potentially relevant in future clinical trials.

Considering these encouraging observations, NRG BR008 (HERO) is a phase III randomized clinical trial expected to launch in the first quarter of 2023 that will include women ≥ 40 years diagnosed with early-stage, low-risk HER2-positive invasive breast cancer (those with pT1N0 receiving chemotherapy or those with clinically < 3 cm node-negative cancer achieving pathological complete responses with neoadjuvant chemotherapy and HER2-targeted therapies). The primary endpoint of the trial is recurrence-free interval amongst all patients surgically treated with breast-conserving surgery and randomized to adjuvant radiation versus no radiation. In addition, relevant oncological outcomes, including ipsilateral breast cancer recurrence, locoregional recurrence, disease-free survival, overall survival, and patient-reported outcomes for pain and fear of recurrence, will comprise secondary objectives [94].

### 4.2. Node-Negative Early-Stage Triple-Negative Breast Cancers

TNBC is a remarkably heterogenous disease entity [95]. Nevertheless, significant progress has recently been made in expanding therapeutic opportunities for both early and advanced stage disease [96,97]. Historically, TNBC has been linked with aggressive disease biology and early locoregional and distant relapses [98]. Hence these cancers are managed aggressively with systemic therapies and adjuvant radiation. Nevertheless, compared to non-TNBC, the magnitude of benefit from adjuvant radiation in TNBC seems limited because in the reported studies, using multivariate analyses, women with TNBC have an increased risk of locoregional relapse, independent of systemic treatments [5,99]. Several retrospective series have shown that adverse clinical outcomes prevail even in small (<2 cm), node-negative TNBC [100,101]; hence it is not surprising that survival gains are evident with the use of adjuvant chemotherapies [102]. Albeit retrospective in nature, data from several study cohorts demonstrate the important observation that a subset of node-negative TNBC not exceeding 1 cm in size experiences exceptionally low rates of locoregional and distant relapses even in the absence of chemotherapy [103,104]. In fact, a patient-level meta-analysis of 12 international cohorts comprising 1835 early-stage chemotherapy naïve TNBC has identified a subset of stage I TNBC with high stromal tumor-infiltrating lymphocytes that display an inherently excellent prognosis, potentially making them suitable candidates for therapeutic de-escalation [105].

Limited studies have addressed the adequacy of locoregional control in small, node-negative TNBC when radiation therapy is omitted from the treatment plan. Eaton et al. queried the Surveillance Epidemiology and End Results database to investigate the influence of radiation after breast-conserving surgery among elderly women (≥70 years) diagnosed with estrogen receptor-negative, T1–2 node-negative breast cancers between 1993–2007. Cumulative incidences of salvage mastectomies (a surrogate for adequacy of local tumor control) and breast cancer-specific deaths were reported for 3432 patients, among whom about 16% were radiation naïve. Their results showed a significantly higher 5-year cumulative incidence of mastectomies (8.3% vs. 4.9%) and breast cancer-specific mortality (24% vs. 10.8%) in the radiation naïve group compared to those that received radiation. However, an exploratory subgroup analysis did find that women ≥ 80 years derived somewhat limited benefit from radiation (mastectomy incidence amongst radiation recipients versus radiation naïve group: 3.4% vs. 6.9%, *p* = 0.05) [106]. Another independent analysis of the National Cancer Database cohort compared overall survival with or without adjuvant radiation after breast-conserving surgery for T1N0M0 TNBC among women ≥ 70 years and revealed a significantly inferior overall survival in the radiation naïve group compared to the group that received adjuvant radiation. Factors associated with adverse outcomes in the radiation naïve group included re-excision for positive margins, tumor size ≥ 2 cm, multiple comorbidities, lower socioeconomic status, and treatment at academic centers [107]. Another study by the same group included data from more than 14,000 non-metastatic pT1–4 node-negative TNBC treated with upfront mastectomy. The authors assessed the factors influencing the use of postmastectomy radiation and showed that pathological tumor size ≤ 2 cm with histologically negative margins, advanced age, treatment at academic centers, and omission of chemotherapy showed a positive association with the omission of adjuvant radiation. Importantly, a significant improvement in overall survival was observed only in pT3 tumors treated with radiation, whereas overall survival was similar in pT1–2 and in pT4 tumors regardless of adjuvant radiation [108].

These retrospective observational data, with their inherent limitations, support a pressing need for prospectively addressing if there is a role for escalating or de-escalation adjuvant radiation in T1–2 node-negative TNBC. In this regard, prospective data is far more limited. Wang et al. performed a multicentre prospective randomized clinical trial to investigate if the addition of radiotherapy improved the clinical outcomes in women (*n* = 681) with stage I–II TNBC treated with mastectomy and adjuvant chemotherapy. Their results showed that the omission of radiation was indeed associated with significantly worse relapse-free survival and overall survival [109].

The first analysis of the LUMINA prospective trial for radiation de-escalation in low-risk luminal A breast cancers underscores the capacity for relevant standardized and validated biomarkers of risk distinction being able to identify a group who can safely avoid radiation and several similar trials are underway in women with ER-positive breast cancers (as shown in Table 1). Given the heterogeneity of TNBC [95] and the fact that clinicopathological factors alone are not sufficient to recapitulate this molecular complexity, biomarker-directed approaches will need to be utilized for patient selection in prospectively designed trials to assess if there indeed exists a group of women with TNBC who can safely avoid radiation therapy.

More recent reports have observed very low rates of local recurrence following lumpectomy in patients who have a complete response to neoadjuvant chemotherapy. The Ontario Clinical Oncology Group is mounting a prospective cohort trial similar to LUMINA where patients with T1–3N0 disease who have had a complete response to neoadjuvant chemotherapy following lumpectomy, including triple-negative disease, will not receive RT and be followed.

## 5. De-Escalation of Adjuvant Locoregional Radiation in Clinically Node-Positive Breast Cancer following Neoadjuvant Chemotherapy

The integration of neoadjuvant chemotherapy into the management of early-stage breast cancer has surged significantly in recent years [110]. Pooled analysis of 33 studies, including 57,531 patients, has demonstrated that axillary pathological complete response (pCR) rates following neoadjuvant chemotherapy with clinically positive axillary nodes vary widely within breast cancer subtypes, with hormone receptor-/HER2+ cancers showing the highest rate (60%) while only 13% of patients with luminal A subtype tumors achieved pCR. The pCR rates for other major subtype definitions are reported as follows: 59% for HER2+, 48% for TNBC, 45% for hormone receptor+/HER2+, 35% for luminal B, and 18% for hormone receptor+/HER2− [111].

An area of much controversy has been the post-neoadjuvant management of patients with clinically positive axillary nodes. Compared to those with residual disease after neoadjuvant chemotherapy, axillary nodal pCR [111] confers a significant survival advantage, with the best prognosis being observed in triple-negative and HER2+ subtypes [112]. This has led to a gradual shift in surgical practice from the routine use of axillary lymph node dissection to less extensive axillary interventions for pathological evaluation, including sentinel lymph node biopsy [113], targeted axillary dissection [114,115], and Marking of the Axilla with Radioactive Iodine (MARI) [116]. The variability of axillary procedures in the post-neoadjuvant setting clearly reflects a current lack of consensus among expert panels on the most accurate axillary staging strategy [3,4,117,118,119].

With regards to regional nodal irradiation, the current guidelines recommend considering its use in patients, particularly those with risk factors, with clinically node-positive axillae, irrespective of the pathological response to neoadjuvant chemotherapy. Nevertheless, there may be patients who achieve pCR in the axillary lymph nodes and who could be potentially considered as candidates for de-escalation of regional nodal irradiation. Much of this speculation is based on retrospective analyses. Barrio and colleagues investigated the rate of nodal recurrence in a series of consecutive patients with clinically node-positive axillae who received neoadjuvant chemotherapy and standardized sentinel lymph node biopsy alone for axillary staging (without further axillary dissection). All 610 patients with clinically node-positive breast cancer received doxorubicin-based neoadjuvant chemotherapy. About 90% of patients (*n* = 555) were rendered node negative; of these, 42% (*n* = 234) were subjected to sentinel lymph node biopsy with the retrieval of up to three sentinel lymph nodes. Though 70% (*n* = 164) of these patients received regional nodal radiation in this cohort, only a single patient developed locoregional recurrence (rate = 0.4%) at a median follow-up of 35 months, supporting the oncological safety of standardized sentinel lymph biopsy alone [120]. Likewise, European Institute of Oncology authors have reported an axillary failure rate of 1.8% when the axillary evaluation was limited to the removal of a single sentinel lymph node after primary chemotherapy in a cohort of patients with clinically node-positive or node-negative axillae. Only 11% of breast-conservation surgery and 38% of mastectomy patients with clinically node-positive axillae received regional nodal irradiation [121].

Haffty and colleagues retrospectively analyzed locoregional recurrence rates among women with T0–T4, N1–N2, M0 breast cancer treated with neoadjuvant chemotherapy and radiation therapy in the ACOSOG Z1071 trial. The decisions about adjuvant radiation in this trial were made by the treating radiation oncologists’ best judgment rather than being prescribed by protocol. The reported overall locoregional recurrence risk was 6% after a mean follow-up of 5.9 years. Subgroup analysis of patients with axillary pCR revealed that omission of postmastectomy radiation and regional nodal radiation after breast-conserving surgery did not adversely influence the locoregional relapse risk [122]. Contrary to these data, some studies have instead reported significantly poor locoregional control with the omission of radiation [123,124]. These estimates are based on retrospective analyses and are possibly susceptible to biases due to confounding factors and selection.

The question of de-escalating regional nodal irradiation after neoadjuvant chemotherapy has been recently addressed in a multicentre Dutch prospective registry cohort (RAPCHEM; BOOG 2010-03) that included 838 patients diagnosed with breast cancers measuring up to 5 cm with 1–3 positive axillary lymph nodes, who received neoadjuvant chemotherapy followed by surgery [125]. The primary endpoint was the 5-year locoregional recurrence rate. As per study protocol, a clinically positive axillary status required the presence of up to three radiologically suspicious axillary nodes with pathological confirmation of metastasis in at least one. In contrast to the ACOSOG Z1071 trial protocol [122], the recommendation for regional nodal irradiation after neoadjuvant chemotherapy was based on three prespecified locoregional recurrence risk categories [(low-risk, ypN0 (i.e., complete pathological response with no residual disease in axillary lymph nodes based on axillary lymph node dissection or sentinel lymph node biopsy); intermediate-risk, ypN1 (i.e., partial pathological response with residual disease in 1–3 axillary lymph nodes based on axillary lymph node dissection); high-risk, ypN2–3 (i.e., residual disease in ≥4 nodes based on axillary dissection)]. For the full study cohort, the 5-year locoregional recurrence rate was 2.2%, supporting the oncological safety of omitting regional nodal irradiation in low- and intermediate-risk groups, i.e., those with pre-treatment clinically positive axillae that downstage to either no residual disease or up to 1–3 positive lymph nodes [125].

Individualization for optimal locoregional management of node-positive patients receiving neoadjuvant chemotherapy is being investigated in two ongoing clinical trials. NSABP B51/Radiation Therapy Oncology Group 1304 is a phase III multicentre randomized clinical trial that is investigating if the addition of regional nodal irradiation to postmastectomy chest wall radiation or whole breast radiation after breast-conserving surgery will significantly reduce the event rate for invasive breast cancer recurrence, in patients diagnosed with breast cancers more than 5 cm in size with up to 3 positive axillary lymph nodes (pathologically confirmed by fine needle aspiration cytology or core biopsy) that convert to pathologically negative axillary nodes following primary chemotherapy. A total of 1636 patients are enrolled. The trial was activated in 2013 and is expected to complete in 2028 (NCT01872975) [126]. Alliance 011202 is a phase III non-inferiority clinical trial in which women with breast cancers more than 5 cm in size with up to 3 positive axillary lymph nodes, who have a residual positive sentinel lymph node following neoadjuvant chemotherapy, are subsequently randomized to axillary lymph node dissection with nodal irradiation or to nodal irradiation alone (NCT01901094). The primary endpoint is invasive breast cancer recurrence-free interval. It is important to note that while both these trials include patients unselected with regards to ER, PR, and HER2 status, responses to neoadjuvant chemotherapies will vary with molecular subtype [111]. Hence incorporation of correlative biomarker studies is imperative to draw the most meaningful conclusions for individualizing critical therapeutic decisions that can be effectively generalized and implemented beyond the setting of this clinical trial.

## 6. Immune Responses in Early Breast Cancer: Ongoing Clinical Trials of Preoperative Radiotherapy and Evidence from Prospective-Retrospective Translational Studies

Tumor-infiltrating lymphocytes (TILs) are populations of mononuclear host immune cells that display phenotypic and functional heterogeneity. A pro-inflammatory, anti-tumoral role is predominantly mediated by CD8+ cytotoxic T cells, natural killer cells, dendritic cells, and M1 macrophages. In contrast, CD4+ regulatory T cells, CD4+ Th2 cells, M2 macrophages, and myeloid-derived suppressor cells promote an immune inhibitory, protumoral milieu [127]. The level of lymphocytic infiltration, as assessed simply and inexpensively by light microscopy on standard hematoxylin and eosin (H&E) stained sections, has evolved as a promising surrogate biomarker of a pre-existing host adaptive immune response portending favorable prognosis and has attained level 1B evidence for clinical utility in early-stage TNBC [128]. Furthermore, stromal TILs have also shown potential for identifying such intrinsically low-risk TNBCs that chemotherapy de-escalation could be considered as a potentially safe choice [129,130]. The clinical relevance of TILs for predicting response to adjuvant [131,132] or neoadjuvant systemic therapies [133,134,135] alone or in combination with immune checkpoint inhibitors is gaining momentum [136,137]. However, the value of immune biomarkers in relation to radiation responses and clinical outcomes in early breast cancer is much less well explored.

Ionizing radiation promotes several alterations in the targeted malignant cells and their associated microenvironmental niche that may impact the immunogenicity of the irradiated tumor. On the one hand, radiation elicits DNA damage leading to immunogenic cell death of the cancer cells, which in turn activates adaptive and innate immune responses that boost anti-tumoral effector functions of cytotoxic T cells. On the other hand, radiation-induced rebound immune suppression is fostered through the recruitment of protumorigenic macrophages, increased expression of immune checkpoints on tumor cells, and TGF-β stimulated accumulation of regulatory FOXP3+ T cells that suppress adaptive immune responses [138,139].

Compared to TNBC, ER+ breast cancers are generally regarded as less immunogenic as they are associated with low levels of lymphocytic infiltrates, immune checkpoint activation, and tumor mutation burden [140]. Hence the immune priming potential of radiation provides at least a theoretical opportunity for switching these immunologically cold tumors to an inflamed phenotype [141] and is being actively investigated in ongoing clinical trials (Table 2). The initial results are available for one of these trials. The SPORT trial (Single Pre-Operative Radiation Therapy for low-risk breast cancer) investigated residual disease burden and immunological responses following single-dose preoperative radiotherapy in women ≥ 60 years with ER+/HER2− T1N0 breast cancers surgically treated with partial mastectomy and sentinel lymph node biopsy. While no complete pathological responses were seen, a partial response was seen in patients undergoing delayed surgery (11–13 weeks) but not in those operated on within 24–72 h after radiation. No significant enrichment in lymphocytic infiltrates was observed at the ablative dose of 20 Gy. No recurrences have been recorded up to 11 months in the follow-up period [142].

Multi-omic profiling has identified an immune-hot subset corresponding to an immunomodulatory subtype of TNBC which is considered most likely to respond to immune checkpoint inhibitor therapy [143]. The comparatively high frequency of this TNBC subtype perhaps explains the relative success of recent trials evaluating the combination of immune checkpoint inhibitors and chemotherapy in the neoadjuvant setting in unselected TNBC, where a pathological complete response is achieved in up to 65% of cases [136,144]. In this context, it remains an outstanding question as to whether radiation-induced immune augmentation could improve therapeutic responses in some TNBCs. Table 2 summarizes the ongoing trials evaluating preoperative radiotherapy alone or in combination with immune checkpoint inhibitors in early-stage disease. Of these, BreastVAX is a phase 1b/2 trial investigating the feasibility and efficacy of combining a single dose infusion of pembrolizumab with radiation boost (delivered as a single fraction of 7 Gy) in patients with operable breast cancers, including TNBC and hormone receptor-positive/HER2 negative tumors [145,146]. The inclusion criteria, however, do not require evaluation of baseline tumoral immune profile. Feasibility and tolerability were evaluated as primary endpoints, and secondary endpoints include pathological complete responses and percentage change in tumor-infiltrating lymphocytes in pre- versus post-treatment samples. The preliminary results have shown major pathological complete responses (<10% viable tumor) in 3 of 9 TNBCs. Compared to the pre-treatment specimens, a significant increase in the density of tumor-infiltrating lymphocytes was seen in the post-treatment tissues of TNBC cases (only). Results of detailed correlative science studies involving digital spatial profiling to identify relevant biomarkers in responsive tumors are pending [147].

**Table 2 cancers-15-01260-t002:** Ongoing clinical trials investigating preoperative radiation therapy in early-stage invasive breast cancer.

Trial Identifier (*n*)	Study Description	Tumor Characteristics	Preoperative Radiation Regime +/− other Therapeutic Agents	Adjuvant Treatments	Endpoints	Prespecified/Exploratory Translational Studies	Estimated Study Completion Year
**Preoperative Radiation (single fraction)**							
NCT01717261 Single Pre-Operative Radiation Therapy (SPORT) for Low-Risk Breast Cancer (SPORT) Phase I (*n* = 13) [142,148]	To investigate the tolerability of a Single Pre-Operative Radiation Therapy (SPORT)	Age ≥ 60 years cT1N0M0 ER+, HER2− unifocal, invasive ductal cancers	Single fraction of preoperative partial breast radiation dose: 20 Gy	Surgery: Early group (within 24–72 h) Late group (11–13 weeks)	Primary: Acute toxicity Secondary: Chronic toxicity and cosmetic outcome, IBTR	Pre/post analysis for Ki-67 and TILs	2020
NCT02482376 Phase II (*n* = 68) [149]	Preoperative single-fraction radiotherapy	Age ≥ 50 years T1N0M0 ER+/HER2− Invasive ductal histology, DCIS Oncotype RS < 18 (for invasive ductal carcinoma)	Stereotactic Body Radiotherapy: Single fraction of 21 Gy.	BCS	Primary: Physician reported cosmetic outcomes Secondary: Patient reported cosmetic outcomes, rates of local control compared to the historic controls	Analysis for pre/post Ki-67 and gene expression analysis. Analysis of cfDNA for assessment of radiation response	2032
NCT03520894 Radiotherapy in Preoperative Setting with CyberKnife for Breast Cancer (ROCK) (*n* = 25) [150]	Preoperative radiotherapy with CyberKnife	Age ≥ 50 years T1N0M0, ER+/PR+ (≥10%)/HER2−, No LVI	Single fraction of 21 Gy	BCS	Primary: Acute skin toxicities Secondary: (3-years) pCR, rate of complete resection with <1 cm margin, LRR, metastasis progression- free survival, cause-specific survival OS, chronic cutaneous and extra-cutaneous toxicities	Radiogenomic analysis using validated signatures. Quantitative immunological analyses using fresh biopsies. IHC-based analysis for pericytes and assessment of vascularization. Serial biochemical analysis of peripheral blood and urine for biomarkers of oxidative stress	2024
NCT02212860 Stereotactic Image-Guided Neoadjuvant Ablative Radiation Then Lumpectomy (SIGNAL 2) (*n* = 139) [151]	Randomized trial to investigate 1 vs. 3 doses of preoperative stereotactic radiation therapy.	Age ≥ 50 years T < 3 cm, node-negative ER+/HER2−	Volumetric modulated arc therapy will be used to deliver: Single fraction of 21 Gy versus 3 fractions of 10 Gy	Surgery after 3 weeks	Primary: Biomarker assessment for immune priming, angiogenesis, hypoxia, proliferation, apoptosis, and invasion. Secondary: Cosmesis, DFS, mastectomy-free survival, and OS	Specified as primary endpoints	2023
**Preoperative Radiotherapy (more than a single fraction)**							
NCT04360330 Study for Selected Early-Stage Breast Cancer (SABER) Phase 1b (*n* = 18) [152]	To determine the most effective dose of preoperative radiation therapy that can be delivered in shorter duration before standard partial mastectomy/axillary surgery	Age ≥ 50 years Unifocal T1N0M0 ER+/PR+/HER2− (Oncotype MammaPrint required)	4 prespecified levels: (35 Gy, 40 Gy, 45 Gy, 50 Gy) in 5 fractions given on non-consecutive days, spanning 2 weeks	Partial mastectomy/axillary surgery 4–6 weeks after preoperative SABER. Standard of care adjuvant systemic therapy	Primary: Establish the most effective preoperative SABER dose. Secondary: Toxicity, pCR, cosmesis, and quality of life	Blood and tissue-based biomarkers Multiparametric MRI studies for assessment of radiation response	2025
NCT04234386 Phase Ib (*n* = 50) [153]	To determine safe and effective dose of pre-operative radiation delivered by FDA approved GammaPod device	Age ≥ 45 years T < 3 cm, N0, ER+/HER2− unifocal, ductal histology, no LVI	Delivery of focussed radiation using GammaPod: 4 prespecified doses: (21 Gy, 24 Gy, 27 Gy, 30 Gy)	BCS	Primary: Establish the most effective single-fraction radiation dose Dose-limiting toxicities Secondary: (5-years) Acute and late toxicities, surgical complications, cosmesis, quality of life, pCR and 5-year IBTR	Not stated	2028
NCT03624478 Phase II (*n* = 25) [154]	Hypofractionated radiotherapy to the whole breast alone before surgery	T0–2, N0 ≥18 years	Hypofractionated radiation therapy daily for 5 days.	Breast-conserving surgery 4–16 weeks after preoperative radiation	Primary: pCR. Secondary: Acute and late toxicities, LRR, distant recurrence, cause-specific survival, DFS, OS	Pre/post-treatment tumor mutation signatures	2022
NCT03043794 Phase II (*n* = 40) [155]	Single fraction Stereotactic Body Radiotherapy to the intact breast		Stereotactic Body Radiation: 21 Gy	Surgery	Primary endpoint: RCB 4–6 weeks after radiation prior to surgery Secondary: Toxicities, local recurrence, cosmesis, quality of life.	Not stated	2026
**Preoperative Radiotherapy and Immune Check Point Inhibitors**							
NCT04454528 Radiation Boost to Enhance Immune Checkpoint Blockade Therapy (BreastVAX) Phase 1b/2 (*n* = 27) [146,147]	To investigate feasibility and efficacy of preoperative pembrolizumab +/− tumor-directed radiotherapy fraction	Age ≥ 18 years T1–2, N0–1M0 TNBC, hormone receptor+/HER2−, Hormone receptor +/− and HER2+	Single dose pembrolizumab +/− Hypofractionated (single fraction of radiation boost: 7 Gy)	Standard of care surgery	Primary: Feasibility of experimental treatment with no delay in surgery. Clinical response (physical exam, breast ultrasound, and histological evaluation)	Comparison of pre/post-treatment immune response on blood and tissue samples	2024
NCT03366844 Phase I/II (*n* = 60) [156]	Preoperative pembrolizumab and radiation boost	First cohort: ER+/HER2− with high-risk features (T1) Second cohort: TNBC T1	Pembrolizumab × single dose followed by second dose of pembrolizumab with radiation boost (24 Gy in 3 fractions)	Surgery and/or chemotherapy (within 8 weeks of enrollment) followed by standard radiation	Primary: Feasibility of experimental treatment with no delay in surgery Secondary: Treatment toxicities, iDFS, pCR	Change in TIL counts	2023
NCT03875573 Neo-CheckRay Phase II (*n* = 147) [157]	Neo-adjuvant chemotherapy combined with stereotactic body radiotherapy +/− durvalumab, +/− oleclumab (Neo-CheckRay) in luminal B breast cancers		Luminal B breast cancer patients randomized to: 1. paclitaxel→ddAC+preoperative radiation boost 2. Arm 1 + durvalumab 3. Arm 2 + antiCD73 antibody	Surgery 2–6 weeks after completion ddAC	Primary: Toxicities, Feasibility of surgery, Pathological evaluation for RCB Secondary: iDFS and cosmetic outcomes	Not stated	2026

Abbreviations: ER, estrogen receptor; HER2, human epidermal growth factor receptor 2; pCR, pathological complete response; RCB, residual cancer burden; DFS, disease-free survival; OS, overall survival; LRR, locoregional recurrence; iDFS, invasive disease-free survival; IBTR, ipsilateral breast tumor recurrence; TILs, tumor-infiltrating lymphocytes; IHC, immunohistochemistry; BCS, breast-conserving surgery; DCIS, ductal carcinoma in situ; RS, recurrence score; cfDNA, cell-free deoxyribonucleic acid; pCR, pathological complete response; ddAC, dose-dense doxorubicin, and cyclophosphamide.

While mature results from these ongoing trials are awaited, archival materials from completed randomized trials linked with long-term follow-up data can provide a valuable resource to investigate the impact of pre-treatment immune cell composition on prognosis and radiation response prediction.

Kovacs et al. [158] investigated the clinical value of stromal TILs on H&E stained sections prepared from pre-treatment primary tumor specimens of patients diagnosed with node-negative, stage I–II early breast cancers who were randomized to breast-conserving surgery with or without whole breast irradiation in the SweBCG91RT clinical trial [159,160]. Their results showed that among patients assigned to the radiation arm, high stromal TILs (≥10%) were positively associated with a significantly lower probability of ipsilateral breast tumor recurrence in multivariate analysis. Patients whose tumors exhibited low stromal TILs (<10%) derived significant benefits from radiation as opposed to those with high stromal TILs, though the interaction test between radiation and TILs was not significant [158]. The authors expanded on their translational study by characterizing CD8 and FOXP3 expressing T lymphocytes by immunohistochemistry. They found that in contrast to immune-depleted tumors (CD8^low^/FOXP3^low^), immune-rich tumors (CD8^high^/FOXP3^low^) showed significantly reduced hazards for ipsilateral breast tumor recurrence or for any recurrence amongst unirradiated patients, perhaps instantiating the antitumoral attributes of the cytotoxic T cells. Additionally, the immune-depleted phenotype appeared to be of benefit from radiation. However, no such advantage was evident in the immune-rich tumors [161]. The relationship between the stromal TIL density and prognostic versus predictive value is rather counterintuitive in the translational studies by Tullberg and colleagues. This may be partly explained by an intrinsically favorable tumor biology indicated by a high stromal TIL density at baseline that translates into satisfactory local control. It is conceivable that these tumors may have an excellent outcome regardless of radiation. Alternatively, or in addition, radiation therapy may kill off activated, proliferating immune cells (a detrimental form of “collateral damage”). On the other hand, tumors with low stromal TILs are perhaps immunologically muted with a higher baseline risk, where radiation therapy appears to be useful in achieving optimal local control by potentially inducing antitumoral immune responses, perhaps through the release of neoantigens from tumor cells killed by radiotherapy (an abscopal effect). 

The value of pre-treatment immune infiltrates has been recently examined in the Canadian MA.20 phase III clinical trial in which women undergoing breast-conserving surgery for T1–2, node-positive or node-negative breast cancer with poor risk features were randomized to standard irradiation with or without regional nodal radiation [162]. The results have shown that both CD8+ and H&E assessed stromal TILs informed favorable clinical outcomes when quantified as a continuous variable. Only CD8+ stromal TILs as a continuous parameter predicted response from regional nodal irradiation [163].

Taking advantage of the randomized design of the Danish Breast Cancer Cooperative Group 82bc clinical trial [164,165], Tramm and colleagues investigated the value of stromal TILs for predicting response from post-mastectomy irradiation. They reported that in the full cohort, high TILs (≥30%) were favorably associated with overall survival and risk of distant metastasis. However, no prognostic value of TILs was found with regard to locoregional relapse risk. High stromal TILs were predictive for benefit in the group randomized to radiation for the endpoint of overall survival. Stratification according to ER status showed that ER-negative tumors with high TILs derived greater benefit from post-mastectomy radiation, whereas no such benefit was observed in ER-negative cases with low TIL counts. The improvement in the locoregional recurrence was independent of the immune infiltration [166].

Building on the abundance of clinical evidence supporting the role of immune biomarkers in risk stratification and guiding decisions for chemotherapy and immune checkpoint inhibitors, analogous data with respect to radiation therapy in early breast cancers is only beginning to emerge from prospective-retrospective studies. Since these trials were not originally designed for subtype-based translational studies, the lack of statistical power remains a major shortcoming in generating consistent results. Hence prospective validation in biomarker-directed studies is imperative. Testing the immune priming potential of radiation in combination with chemotherapy and/or immune checkpoint inhibitors provides an ideal opportunity to induce immune modulation in breast cancers which are largely considered to be poorly immunogenic. It is expected that accompanying, preplanned correlative studies will allow for an in-depth assessment of immunological responses (or lack thereof) in the primary tumor and draining lymph nodes that can inform future definitive clinical trials.

## 7. Biomarkers to Guide Adjuvant Radiation Decisions

Over the years, several groups have invested significant efforts to develop radiation-specific genomic classifiers for prognostication of locoregional relapse risk and prediction of response to radiation therapy. These classifiers have been reviewed in detail in previous publications [167,168], and those with the potential for clinical development are summarized in Table 3. To date, none of these classifiers have progressed to stages of analytical and clinical validity which is critical before these genomic assays can be tested for their clinical utility in phase III randomized trials [169,170,171].

Here we will focus on liquid biopsy-based approaches and review the recent investigations into the role of disseminated tumor cells and circulating tumor cells as prognostic biomarkers for locoregional relapse risk.

### 7.1. Disseminated Tumor Cells

Disseminated tumor cells (DTCs) are isolated cancer cells that, upon physical detachment from the primary tumor, escape the circulation, extravasate into distant sites such as bone marrow, and are capable of survival in a hostile host niche, reversible quiescence, and therapeutic resistance. DTCs detected via bone marrow aspiration are found in approximately 40% of women with stage I–III breast cancers who do not have any clinical or histological evidence of overt metastatic disease at initial presentation. A substantial body of evidence from clinical studies has demonstrated that compared to patients without DTCs, those with DTC positivity have features of aggressive tumor biology, including larger tumor size, higher grade, axillary lymph node metastasis, estrogen/progesterone receptor negativity, and HER2 positivity [182,183]. An earlier pooled analysis of 9 studies comprising 4703 patients with operable breast cancer (enrolled before 2002) provided evidence for a strong association of DTCs with significant adverse outcomes [182]. These findings have been further confirmed in a recently published patient-level meta-analysis comprising 10,307 early breast cancer patients from 11 centers with a median follow-up of 7.6 years [183].

Only a few studies have investigated the impact of bone marrow occult metastasis on locoregional relapses, showing either no association [183,184,185] or a significantly increased risk of locoregional failures. In a single centre prospective cohort of more than 3000 stage I–III treatment naïve breast cancers, Hartkopf and colleagues demonstrated bone marrow DTCs in 24% of patients at the time of initial surgical intervention. DTC positivity was independently associated with locoregional failures. Their results further revealed that, of the available biopsy samples from patients with isolated local relapses, the 55 subjected to a repeat bone marrow aspiration showed a DTC detection rate of 35% [186]. Bidard et al. investigated the relationship between locoregional relapse-free survival and bone marrow DTCs in a prospective cohort of 621 patients from Institute Curie’s Breast Cancer Micrometastasis Project for a median follow-up of 4.6 years. In this cohort, 15% of patients had detectable DTCs in the bone marrow. Overall, 18/621 patients experienced a locoregional relapse, among whom 44% had evidence of DTCs at their initial evaluation. Amongst patients with DTC positivity, a longer locoregional relapse-free survival was observed in patients who received endocrine therapy and radiation to supraclavicular/internal mammary lymph nodes [187]. These results remained consistent at an updated median follow-up of 11.7 years, where there was a 10-year locoregional relapse rate of 20% in patients with DTC-positive status compared to 10% in those without, supporting the capacity of DTCs as a biomarker predictive of benefit from regional nodal irradiation [188]. The biological basis of locoregional relapse in patients with bone marrow micrometastasis is not completely understood. However, preclinical studies using mouse models suggest that DTCs may transition into circulating tumor cells, a fraction of which have the potential to re-colonize the primary tumor site [189]. It is conceivable that irradiating regional lymph nodes in patients with DTC-positive status may eradicate subclinical micrometastases and may serve as a candidate predictive biomarker for optimizing patient selection for regional nodal irradiation. This may be potentially relevant in the context of selecting patients who may benefit most from irradiation of regional lymph nodes [162,190,191,192].

### 7.2. Circulating Tumor Cells

Circulating tumor cells (CTCs) are occult malignant cells that exit from the primary tumor into the circulation and are associated with enhanced metastatic potential [193]. When examined prior to any treatment (neoadjuvant chemotherapy or upfront surgery) by utilizing CellSearch^®^, an FDA-approved standardized assay, the prevalence of CTCs has been found to be 25% in an international meta-analysis including 2156 patients from 21 studies. After eliminating T4 tumors from analysis, CTC positivity did not have a statistically prominent association with clinicopathological factors or pathological complete response. However, CTC presence prior to initiation of neoadjuvant therapy was indicative of shortened disease-free survival, overall survival, and locoregional relapse-free interval in univariate and multivariate analyses. Moreover, the inclusion of baseline CTC counts significantly improved the prognostic capacity of the clinicopathological model [194].

Goodman and colleagues have reported on the association between CTCs and response to adjuvant radiation in patients with pT1–T2, N0–1 early breast cancer utilizing patients’ clinical and CTC data from the National Cancer Database (*n* = 1697) and validated their findings in a cohort from the German SUCCESS trial [195,196] (*n* = 1516). CTC positivity was associated with the benefit from adjuvant radiation with a significant increase in overall survival in the National Cancer Database cohort and in disease-free-, overall, and locoregional relapse-free survival in the SUCCESS cohort. In addition, an improvement in overall survival was seen in CTC+ patients undergoing breast-conserving surgery with standard adjuvant radiation but not in CTC- patients. When stratified by CTC status, the benefit of radiation was not evident in patients treated with mastectomy. These results should be interpreted carefully in light of the existing evidence [5,197], as adjuvant radiation was not the randomization criteria in either of the evaluated cohorts. Nevertheless, these encouraging results support value for CTCs as a potential biomarker for guiding radiotherapy decisions, requiring prospective validation to analyze the benefit of adjuvant radiation therapy in low-risk patients with CTC positivity who otherwise might otherwise be considered for radiation omission.

BreastImmune03 is a randomized phase II clinical trial designed to assess the clinical benefit of post-surgery adjuvant radiotherapy + immunotherapy with nivolumab + ipilimumab, versus radiotherapy + capecitabine in TNBC patients with residual disease after neoadjuvant chemotherapy. Evaluation of CTC will be performed as a secondary outcome measure for immune monitoring at cycles 1, 2, 5, and 2 years post-randomization or in the event of a relapse [198].

## 8. Summary

Adjuvant radiotherapy is an integral component of early breast cancer management, with proven efficacy for preventing locoregional and distant failures. Over the years, traditional whole breast irradiation approaches have evolved considerably, such that the less intensive option of whole breast hypofractionated radiation has now become the preferred standard, yielding improved compliance, cosmetic outcomes, and quality of life. More recent data have shown the comparable efficacy and safety of an ultra-hypofractionation regimen that is delivered as five fractions in less than a week. Equivalence of accelerated partial breast radiation delivered by external beam has been demonstrated in several clinical trials and endorsed for women with tumors with favorable biology, and together with ultra-hypofractionation, may be an attractive option in resource-restricted regions. Investigations for further de-intensifying radiation schedules using ultra-accelerated partial breast radiation as a single fraction are being planned to be tested against accelerated partial breast radiation + endocrine therapy [199,200].

The encouraging results of the LUMINA trial support the safe omission of adjuvant radiation following breast-conserving surgery when selection criteria are strictly limited to low-risk cancers (T1N0, grade 1 or 2) with luminal A phenotype with a Ki-67 index of ≤ 13.25%. This and the other ongoing trials of radiation omission underscore the significance of biomarker-driven risk stratification for critical decisions involving radiation de-escalation. These trials are a step forward in personalizing options for radiation, the integration of which into clinical practice has lagged behind analogous de-escalation protocols in systemic therapy.

Recognizing the immune-modulatory potential of radiation, the ongoing clinical trials of pre-operative radiotherapy will provide opportunities to investigate combinatorial therapies in early-stage settings. Nevertheless, critical to the success of immune priming approaches will be the understanding of the biological interactions of host immunity with key factors that influence immune-modulating properties of radiotherapy, such as radiation dose, quality, fractionation schedules, and sequence of the therapies [201].

Ultimately, the actively evolving scientific understanding of breast cancer biology is driving clinical trials that are providing radiation oncologists and women with breast cancer with the information they need to make personalized choices that both protect them from recurrences and from unwarranted treatment morbidity. In view of the evolving evidence, therapeutic strategies incorporating tumor and patient characteristics, as well as patient preferences, should be discussed in a multidisciplinary tumor board to tailor the treatment for the patient [10].

## Figures and Tables

**Table 1 cancers-15-01260-t001:** Ongoing clinical trials for the omission of adjuvant radiation therapy in molecularly defined low-risk estrogen receptor-positive breast cancers.

Trial Name (NCT ID) & Completion Year	Trial/Study Design (*n*)	Eligibility Criteria							Outcome Measures
		Age (yr)	Pathological Stage	Grade	Receptor Status by IHC	Surgery	Margin Status (mm)	IHC/Genomic Classifier	
**Trials investigating omission of whole breast irradiation in node-negative breast cancers**									
LUMINA (NCT01791829) [69] 2024	Prospective, single arm, observational study (*n* = 501)	≥55	Stage 1 (pT1N0M0)	1–2	ER+/PR+/HER2−	BCS+ SLNB or axillary dissection	≥1	Molecularly defined luminal A by IHC ^1^: ER ≥ 1%, PR > 20%, HER2− (by IHC or in situ hybridization) and Ki-67 ≤ 13.25%	Primary: 5-year ipsilateral invasive breast cancer or DCIS recurrence Secondary: Contralateral breast cancer, RFS based on any recurrence, DFS based on any recurrence, second cancer or death, and OS
PRECISION (NCT02653755) [70] 2026	Phase II prospective cohort study (*n* = 690)	50–75	Stage I (pT1N0M0) (Axillary nodes with isolated tumor cells permitted)	1–2	ER+/PR+/HER2−	BCS+ SLNB or axillary dissection	No ink on tumor or re-excision with no residual disease	Prosigna ROR	Primary: 5-year risk of ipsilateral LRR Secondary: 5-year risk of any recurrence, DFS, and OS
IDEA (NCT02400190) [71] 2026	Prospective, single-arm observational study (*n* = 202)	50–69	Stage I (pT1N0M0) (Axillary nodes with isolated tumor cells permitted)	N/A	ER+/PR+/HER2−	BCS+ SLNB or SLNB→axillary dissection or axillary dissection	≥2	Oncotype Dx RS ≤ 18	Primary: 5-year LRR Secondary (10 years): Recurrence pattern, subsequent therapy for local recurrences, OS, and BCSS
PRIMETIME (ISRCTN: 41579286) [72] 2027	Case-cohort, prospective study (*n* = 1500)	≤60	Stage I (pT1N0M0)	1–2	ER+/PR+/HER2− ^2^	BCS+ SLNB	≥1	IHC4+C	Primary: 5-year IBTR
DEBRA NCT04852887 [73] 2041	Phase III, multicenter randomized trial (*n* = 1670)	50–70	Stage I (pT1N0M0) (Patients with pathologic staging of pN0(i+) or pN0(mol+) are not permitted	N/A	ER+/PR+/HER2−	BCS→ WBI + ET vs. BCS→ET	No ink on tumor or re-excision with no residual disease	N/A	Primary: 5-year invasive or non-invasive IBTR Secondary: Percentage of women with an intact index breast, any invasive IBTR, any breast cancer recurrence at a local, regional, or distant site, recurrence or a secondary primary cancer and death
EXPERT (NCT02889874) [74] 2023	Randomized phase III clinical trial (*n* = 1167)	≥50	Stage I (pT1N0M0)	1–2	ER+/PR+/HER2−	BCS+ SLNB or axillary dissection	No ink on margin	PAM50 Luminal A and Prosigna ROR ≤ 60	Primary: 10-year risk of ipsilateral local recurrence Secondary: 10-year LRR, distant recurrence, DFS including DCIS, iDFS, OS, rates of salvage RT or mastectomy, and quality of life-related endpoints such as convenience of care and fear for recurrence
The DBCG RT Natural Trial (NCT03646955) [75] 2035	Randomized phase III clinical trial (*n* = 926)	≥60	Stage I (pT1N0M0)	1–2	≥10% ER+/HER2−	BCS+ SLNB or axillary dissection	≥2	N/A	Primary: 10-year-invasive local recurrence in ipsilateral breast. Secondary: (10-year) Regional recurrence, distant recurrence, and death.
**Trials investigating omission of regional nodal irradiation in node-positive/negative breast cancers**									
CCTG MA.39 (TAILOR RT) (NCT03488693) [76] 2027	Randomized phase III, clinical trial (*n* = 2140)	≥40	Stage 1 T1–3N0–1	N/A	ER+/HER2− (Local testing)	BCS or mastectomy + SLNB and/or axillary dissection	≥1	Oncotype Dx RS < 18	Primary: Breast cancer recurrence-free interval Secondary: DFS, LRR, OS, breast cancer mortality, distant recurrence, toxicity, arm volume, and mobility assessments, patient reported outcomes and cost effectiveness

^1^ IHC assay performed in local laboratories as per American Society of Clinical Oncology Guidelines. ^2^ As per local practice. Abbreviations: ER, estrogen receptor; PR, progesterone receptor; HER2, human epidermal growth factor receptor 2; BCS, breast-conserving surgery; SLNB, sentinel lymph node biopsy; ROR score, risk of recurrence score; RS, recurrence score; LRR, local regional recurrence; DCIS, ductal carcinoma in situ; RFS, relapse-free survival; DFS, disease-free survival; IBTR, ipsilateral breast tumor recurrence; iDFS, invasive disease-free survival; OS, overall survival; BCSS, breast cancer-specific survival; WBI, whole breast irradiation; ET, endocrine therapy; IHC, immunohistochemistry; IHC4+C, immunohistochemistry 4+ clinical.

**Table 3 cancers-15-01260-t003:** Radiation Specific Genomic Classifiers.

Genomic Classifier	Description of the Classifier	Breast Cancer Cohort/Trial Characteristics	Prognostic Value	Predictive Value
Radiation Sensitivity Index (RSI) [172,173,174]	- Systems biology-based pan cancer radiosensitivity classifier of 10 hub genes that are involved in regulating radiation signaling pathways. - Developed by modeling the survival fraction of 48 human cancer cell lines at 2 Gy as a measure of cellular radiation responsiveness such that RSI index is directly proportional to radioresistance. - Validated in rectal, esophageal, and head and neck cancers treated with chemoradiation. - RSI is measured as a continuous score and categorized into 3 categories as: (a) RSI radioresistant subtype: Top 25th percentile of RSI scores, (b) RSI radiosensitive subtype: Lower 25th percentile of RSI scores, (c) RSI-intermediate subtype: RSI scores between the 25th–75th percentile	FIRST PUBLICATION COHORTS [173] (I) Karolinska prospective cohort: segmentectomy/mastectomy +RT (*n* = 77); mastectomy only (*n* = 82) (II) Erasmus cohort: BCS+RT (*n* = 219) MRM (*n* = 67) SECOND PUBLICATION COHORTS [174]: 4 Dutch + 1 French cohorts (*n* = 343) BCS+SLNB/Axillary dissection→ WBI+/− RNI (Integration of RSI index with breast cancer molecular subtype)	RSI is a radiation-specific signature that has shown prognostic value in RT treated group but not in the no-RT group. FIRST PUBLICATION [173] Karolinska cohort: Compared to radioresistant patients, radiosensitive patients had improved 5-year RFS. Erasmus cohort: Compared to radioresistant patients, radiosensitive patients had improved 5-year DMFS. Multivariate analysis: Independent prognostic variable associated with outcome in RT-treated patients in both cohorts and in RT-treated ER+ subset in the Erasmus cohort. SECOND PUBLICATION [174]: -RSI index was not prognostic in the full cohort. - In patients with triple-negative subtype, RSI-resistant tumors were associated with higher risk of local relapse compared to those with RSI-sensitive/intermediate categories.	No
Radiation Sensitivity Signature (RSS) and Immune Signature (IMS) [175]	Gene signatures are based on intrinsic radiation sensitivity and antitumor immunity. RSS: 34 gene classifier was derived from MSigDB. IMS: comprised of 119 genes involved in antigen presentation and processing pathways curated from the Immunology Database and Analysis Portal. Four genes (*ADRM1*, *MICB*, *PSMD13*, *RFXANK*) showed significant interaction with radiotherapy. Immune-effective: IMS score > −3.8 Immune defective: IMSscore < −3.8	Model training cohort for RSS: GSE30682 cohort (*n* = 343) treated with BCS+RT. Endpoint: LRFS Model training cohort for IMS: E-TABM-158 RT: *n* = 66, No RT: *n* = 63. Endpoint: DSS Validation of ISS and IMS: METABRIC cohort (*n* = 1981)	In the METABRIC cohort: For radiation-sensitive group, patients who received radiotherapy experienced an improved DSS than those who did not.For immune-effective group, patients treated with radiotherapy had significantly better DSS compared with those without radiation therapy. Combined ISS and IMS were validated in the METABRIC cohort. Patients were categorized into four groups: -Concordant group: immune-sensitive/immune effective group treated with radiation had significantly better DSS. -Concordant group: immune-resistant/immune defective treated with radiation had significantly poor DSS. No significant prognostic associations were found in the two discordant groups.	When evaluated independently, both RSS and IMS predicted benefit from radiation in the RT-treated cohort for the endpoint of DSS. Integration of RSS and IMS stratified patients into groups. Benefit from radiation is seen in the radiation-sensitive/immune effective group treated with radiotherapy. Radiation resistant/immune defective group did not derive benefit from radiation.
DBCG-RT Profile [176]	*7* gene classifiers (*HLA-DQA*, *RGS1, DNALI1*, *hCG2023290*, *IGKC*, *OR8G2*, and *ADH1B*) developed on the fresh frozen tissues from the training cohort. The classifier stratified the training cohort into high- and low-risk groups for locoregional relapse. The final signature consisted of 4 genes (*IGKC*, *RGS1*, *ADH1B*, and *DNALI1*) as the remaining 3 genes failed quality control during transfer to formalin fixed paraffin embedded tissues.	DBCG82b/c randomized clinical trial: 3083 women (<70 years) with high-risk disease randomized PMRT+RNI or not. All post-menopausal women received tamoxifen (82c) and premenopausal women (82b) received CMF. Training set = 191 Validation set: 112	Prognostic value was assessed in the non-irradiated group of training set who received systemic treatments and stratified the population into two groups: low LRR risk and high LRR risk, which demonstrated low- and high-risk for locoregional failures, respectively. Multivariate analysis: DBCG-RT profile provided independent prognostic information for locoregional relapse risk.	Predictive value was demonstrated in both training and validation cohorts. DBCG-RT profile predicted benefit from PMRT in patients classified as high-risk-LRR. No benefit was derived from PMRT in DBCG-RT low-risk category.
Radiation Sensitivity Signature (RSS) [177]	Clonogenic survival assays were performed on a panel of 16 breast cancer cell lines to identify the surviving fraction at 2 Gy, which represents the intrinsic radiosensitivity or radioresistance of the breast cancer cells. The classifier was trained and cross-validated from 147 to 51 genes enriched in cell cycle, DNA damage, and DNA repair pathways. The classifier was independent of breast cancer molecular subtypes.	Training cohort (*n* = 343) treated with BCS + RT for which locoregional recurrence data were available. Validation cohort (*n* = 228) treated with mastectomy or BCT and radiation if indicated.	RSS provided prognostic information for overall survival of locoregional recurrences and stratifies patients unlikely to develop local recurrence after radiation from those at high risk of recurrence despite receiving standard radiation.	No
Adjuvant Radiotherapy Intensification Classifier (ARTIC) [178]	Clinicogenomic classifier comprising of 27 genes and age.	Training cohort: 3 publicly available data sets with gene expression data: Servant (*n* = 343) van de Vijver (*n* = 228) Lund fresh frozen (*n* = 102) Validation cohort: SweBCG91-RT phase III trial cohort (*n* = 748) in which patients were treated with BCS with or without radiation.	ARTIC provided prognostic information for locoregional recurrence in both the treatment arms (with or without radiation).	Patients with low ARTIC scores derived significant benefits from radiation for the endpoint of locoregional recurrence compared to patients with high ARTIC scores who gained less from radiation.
Profile for the Omission of Local Adjuvant Radiotherapy (POLAR) [179]	Transcriptome-wide profiling of tumors was performed using the Affymetrix Human Exon 1.0 ST microarray. A 16-gene signature (proliferation and immune response) was trained in the training set of patients who did not receive radiation.	SweBCG91-RT cohort was divided into a training set (*n* = 243) and validation set (*n* = 354)	Tumors with POLAR low-risk had a 10-year locoregional recurrence rate of 7% in the absence of radiation. POLAR high-risk had a significantly decreased risk of locoregional recurrence when treated with radiation.	Independent external validation for predictive performance was performed in 623 patients from three randomized clinical trials (SweBCG91-RT, *n* = 354; Scottish Conservation Trial; *n* = 137 and trial from Princess Margaret Hospital, Canada, *n* = 132) High POLAR score was predictive of benefits from radiation with significant reduction in the local recurrence rate [180,181]

Abbreviations: MRM, modified radical mastectomy; BCT, breast-conserving therapy; RT, radiation therapy; SLNB, sentinel lymph node biopsy; RFS, relapse-free survival; PMRT, post-masectomy radiation therapy; RNI, regional nodal irradiation; DMFS, distant metastasis-free survival; LRR, locoregional recurrence; DFS, disease-specific survival; CMF, cyclophosphamide methotrexate fluorouracil.

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
