# Peer review of "Recent Advances in Optimizing Radiation Therapy Decisions in Early Invasive Breast Cancer"

_cancers, 2023, doi:10.3390/cancers15041260_

Round 1
Reviewer 1 Report
I thank the authors for the opportunity to review this article. It is a very comprehensive work on the very interesting aspect of optimising treatment in early breast cancer. The authors provide an excellent review of the existing evidence and ongoing studies aimed at finding subgroups of patients in whom radiotherapy can be omitted. The main studies that support this attitude are adequately reviewed, included and analysed, extracting the most relevant aspects from them. Ongoing studies include exhaustively the evidence to come. The included tables are informative and not redundant with the text, and the bibliography is up-to-date. The conclusion of the paper supports the authors' initial assumption about the possibility and/or necessity of omitting radiotherapy in selected subgroups of women with early breast cancer.
However, for the sake of objectivity in such a sensitive subject, the authors should perhaps consider including some aspect that has not been included and that could provide other perspectives on the subject under analysis:
- 1.- The authors refer to a hypothetical toxicity (cardiotoxicity) induced by radiotherapy, providing bibliographic citations to support this. However, it would also be interesting for the authors to consider the work of Holm Milo et al. 2021 (doi: 10.1016/j.radonc.2021.01.010), Yit et al. 2022 (DOI: 10.5114/wo.2022.115676) or Overgaard et al. 2022 (DOI: 10.1016/j.radonc.2022.03.008) which, to a certain extent, contradict the initial hypothesis.
- 2.- The authors justify the omission of radiotherapy in early stage patients on the lack of impact on overall survival despite the increase in loco-regional relapses, as demonstrated by PRIME II and similar studies. Perhaps the authors should also assess the impact of locoregional relapse episodes on the quality of life of these patients and the associated costs in terms of additional tests, biopsies, surgery and further treatment.
- 3.- The authors justify the omission of radiotherapy in the administration of systemic treatment to all patients. It would perhaps be appropriate for the authors to analyse the need for hormonal treatment in all patients with early low-risk breast tumours, the number of patients to be treated in order to see benefit (Staley et al. 2012 10.1002/14651858.CD007847.pub2, Rourke et al 2008 DOI: 10.1200/jco.2008.26.15_suppl.17547, Goodwin et al 2009 DOI: 10.1016/j.breast.2009.04.003, Smith et al 2006 DOI: 10.1093/jnci/djjj186) Likewise, it would be desirable for the authors to also analyse patient adherence to sustained hormonal treatment and the pros and cons of this versus limited radiotherapy in terms of quality of life in patients with early breast tumours (Yussof et al. 2022 doi:10.1016/j.breast.2022.01.012, Peddie et al 2019 DOI: 10.1016/j.breast.2021.05.005, Robinson et al 2018 DOI: 10.1016/j.clon.2017.10.015).
- The authors analyse in depth in Table 1 the studies underway with the aim of omitting radiotherapy, with the particularity that all of them contemplate the administration of hormone treatment. It would be interesting for the authors to also consider and discuss those studies underway that aim to omit hormone treatment as opposed to radiotherapy in low-risk tumours, such as the TOP-1, NATURAL and EUROPA trials: https://www.boogstudycenter.nl/studie/283/top-1.html, https://clinicaltrial.gov/ct2/show/NCT03646955, https://clinicaltrial.gov/ct2/show/NCT04134598.
- 5.- The authors analyse the lack of impact on survival of radiotherapy, both in patients with low-risk tumours and in those with unfavourable tumour subtypes. It would be interesting if the authors could also comment on the results observed by Herskovic et al 2018 (DOI: 10.1016/j.clbc.2018.02.006), Bazan et al 2021 (DOI: 10.1038/s41523-021-00242-8), Wu et al 2019 (DOI: 10.1186/s13014-019-1394-x), Haque et al. 2019 (DOI: 10.1111/tbj.13443) that show, in the analysis of large databases, benefit not only in locoregional control but also in survival, with the addition of radiotherapy in early tumours not only of unfavourable subtypes but also in patients considered as low or very low risk.
All in all, I think this is a great work but to be complete it should also include evidence to support limited local treatment versus generalised systemic treatment in patients with early breast cancer.

Reviewer 2 Report
It is a very complete work concerning the correct use of radiotherapy in the treatment of breast cancer.
The first part of the work (and the Abstract) should be remodeled from a perspective of greater scientific objectivity with respect to the role of radiotherapy which is not only harmful due to the side effects or expensive due to the costs of the machines that are used, but certainly useful and necessary in the therapeutic path of patients with breast cancer, if shared and tailored indications are followed with respect to the patient herself.
I would add in the introduction that the patients' quality of life has now absolutely improved during radiotherapy thanks to hypofractionation regimens and techniques that do not involve particular burdens on the duration and results of the treatments. On the contrary, it must be considered that, even if in some of the studies reported radiotherapy does not give significant advantages on survival, it always has an important role on local disease control and local recurrence in turn has a very significant impact on quality of life.
I have added a bibliographic note by Italian authors, regarding the age as a discriminating factor in the choice of treatment.
The English is very fluent, the work is complete as the main evidence in this area has been addressed.

Reviewer 3 Report
Overall, a well written review of studies to optimize the role of radiation therapy in early stage breast cancer.
Nonetheless, the authors do not appear to address the early and delayed adverse effects that can occur when partial breast, whole breast, and/or nodal radiotherapy is administered, especially when delivered before or concurrent with surgery and/or systemic therapy.
This needs to be addressed, especially the wound healing complications that can occur, and how optimizing radiation therapy needs to factor in the adverse event risk when determining if/when/how much radiation should be administered.
Round 2
Reviewer 3 Report
Good response to reviewer suggestions